# Micro RNAs and Circular RNAs in Different Forms of Otitis Media

**DOI:** 10.3390/ijms24076752

**Published:** 2023-04-04

**Authors:** Michal Kotowski, Paulina Adamczyk, Jaroslaw Szydlowski

**Affiliations:** Department of Pediatric Otolaryngology, Poznan University of Medical Sciences, 60-572 Poznan, Poland

**Keywords:** microRNA, miRNA, circRNA, acute otits media, otits media with effusion, chronic otitis media, cholesteatoma

## Abstract

The aim of this comprehensive review was to present the current knowledge on the role of microRNAs (miRNAs) in acute, recurrent, and chronic forms of otitis media. Special attention was focused on cholesteatoma of the middle ear. MicroRNAs modulate gene expression, which, in turn, influences the development and likelihood of the recurrence of acute and aggressive chronic middle ear inflammatory processes. Moreover, this study discusses the modulating role of a specific subgroup of noncoding RNA, circular RNA (circRNA). Recognizing the precise potential pathways and the mechanisms of their function may contribute to a better understanding of the molecular bases of middle ear diseases and identifying novel methods for treating this demanding pathology. Articles published between 2009 and 2022 were used in this analysis. In this review, we provide a complete overview of the latest progress in identifying the role and mechanisms of particular miRNAs and circRNAs in acute, recurrent and chronic forms of otitis media.

## 1. Introduction

Acute otitis media (AOM) is an inflammatory process of the middle ear that is caused by viral or bacterial infection. It is a common disease, especially amongst children, whose middle ear anatomy is different from adults, making them more susceptible to this condition [1]. For the same reason, in this group of patients the recurrence is more often observed and recurrent otitis media (rAOM) with the persistent presence of ear effusion eventually can lead to chronic otitis media (COM) development [2]. Otitis media with effusion (OME) is characterized by a presence of nonpurulent fluid behind the tympanic membrane without any symptoms of acute infection and often is a consequence of AOM or rAOM. Effusion, if it persists for longer periods, may lead to damage to the middle ear and cause conductive hearing loss which is especially important in pre-school children where it can induce speech development delay [3]. The evolution of COM is mediated by cytokines such as IL-1 and TNF-α that are secreted in response to bacterial infection during an AOM episode and trigger an inflammation response which in effect leads to epithelium remodeling [4]. However, the exact molecular mechanisms of COM formation are still not fully understood. Perhaps, miRNAs that control gene expression at the post-transcriptional stage might be involved in this process.

Acquired middle ear cholesteatoma (AMEC) may be a consequence of OME and due to the recurrence rates its treatment remains a real challenge. AMEC ia a solid mass composed of keratinizing squamous epithelium in the middle ear but with the potential to expand to the surrounding structures. Although considered benign, it is characterized by expanding growth that can destroy the bony structures of the middle ear and eventually lead to many serious conditions such as hearing impairment, vestibular dysfunction, facial nerve palsy, or severe intracranial complications [5]. The pathogenesis of chronic otitis media with cholesteatoma (COMC) is multifactorial and still not fully understood. There are four main theories for explaining cholesteatoma formation, which are based on one of the following: invagination, immigration, squamous metaplasia, or basal cell hyperplasia. The invagination theory includes the accumulation of hyperproliferating epithelial keratinocytes with inhibited apoptosis in a progressive retraction pocket, leading to the formation of cholesteatoma [6]. However, the exact mechanisms underlying cholesteatoma development remain unknown [7]. Surgical resection remains the only method of treatment [8]. This provides a strong incentive for researchers to seek an explanation for cholesteatoma formation at the molecular level, which could lead to a paradigm shift in treatment.

MicroRNAs (miRNAs) are small, noncoding molecules that modulate gene expression at the post-transcriptional stage; however, their exact role in the pathogenesis of human diseases remains unknown. Their influence on the human genome is multifactorial—one miRNA can modulate the expression of many genes, and a single gene can be affected by multiple miRNAs [9]. What is known is that miRNAs regulate around 60% of the protein genes in the human genome, making them the most promising group of interest for molecular research [10].

Many studies have investigated the role of miRNAs in the pathogenesis of various diseases, demonstrating that miRNAs actively partake in many vital processes, such as proliferation, maturation, differentiation, apoptosis, and angiogenesis [11,12]. As almost half of miRNA transcripts are located in regions that are affected during carcinogenesis, it is clear that they may play a pivotal role in the development or transformation to malignancy [13]. MiRNAs whose alteration leads to cancer development are commonly called oncomirs; they were first described in cases of chronic lymphocytic leukemia (CLL) where the expression of two miRNAs (miR-15a and miR-16-1) was upregulated, affecting the expression of antiapoptotic B-cell lymphoma 2 (Bcl2) protein [14]. MicroRNAs can function as oncogenes or tumor suppressors; they can also perform both functions simultaneously, as with miR-26, which is an oncogene in cases of glioma and glioblastoma, inhibiting PTEN (a protein regulating the Akt pathway), in addition to acting as a suppressor in other cancers, such as breast cancer, thyroid cancer, and hepatocellular cancer [15,16].

Recent evidence shows that miRNAs have great diagnostic and therapeutic potential, as they are shown to be more stable than mRNAs and can be easily collected and analyzed. They are present in human plasma, serum, saliva, and urine, which not only makes them excellent non-invasive candidates to be used as biomarkers for diagnosis, prognosis, and disease monitoring, but also shows their potential to become future targets for pharmacological treatment [17,18]. The discovery of microRNA-regulated carcinogenesis has led to the development of anti-miRNAs called antagomirs, which may directly exert a therapeutic effect by blocking oncomirs [19].

Some studies have reported that miRNAs might function as ligands transmitting signals to the cell surface receptors, such as in the case of miR-21, which binds to TLRs (Toll-like receptors) in the immune cells, activating the inflammation response but also—in some cases—tumor growth [20]. Inducing the immune response via microRNA-regulated pathways, as later presented in this paper, plays an important role in the development of middle ear pathologies. The most recent studies have focused on a specific subgroup of noncoding RNA—circular RNA (circRNA). These molecules may play a specific modulating role acting as a competing endogenous RNA [21,22].

The aim of this paper is to present a complete overview of the latest progress on the role of miRNA and circRNA in AOM, rAOM, OME, and AMEC, based on a comprehensive literature review.

## 2. Literature Review

This review identifies and summarizes the literature concerning the role of miRNA and circRNA in middle ear pathologies that were published between 2009 and 2022 in the PubMed, Scopus, and Web of Science databases. Each database was searched using the following combination of terms: ‘miRNA’ and ‘otitis media’, ‘circRNA’ and ‘otitis media’, ‘miRNA’ and ‘cholesteatoma’, and ‘circRNA’ and ‘cholesteatoma’. The initial search results were identified and a comparison between the search results in different databases was conducted. The same article in different databases was treated as one position. The inclusion criteria regarding the search results were established as English language original research with the full text available. The exclusion criteria included as follows: studies conducted on non-human tissues, non-English language papers, papers that did not contain original research, or those for which only an abstract was available. The inclusion and exclusion criteria were all defined prior to the search. All articles which met the aforementioned criteria were screened for their applicability based on the full text content. No studies were excluded on the basis of poor-quality methods or nonsubstantiated results. The final research material consisting of 27 articles was analysed and presented in this study (Table 1).

Scientific evidence supporting the involvement of several miRNAs and circRNAs in AOM, rAOM, COME, and cholesteatoma formation are summarized below. Their most important potential regulatory mechanisms, pathways, and clinical outcomes are presented in Table 2 (organized by the type of RNA).

### 2.1. Noncoding RNA in Acute, Recurrent, and Chronic Otitis Media with Effusion

There is a paucity of studies focusing on the presence of miRNAs in the middle ear and the assessment of the impact in the development of acute and chronic inflammation.

A study conducted by Song et al. was the first to describe the presence of miRNA in human middle ear epithelial cells (HMECs) after a 2 h exposition to lipopolysaccharide (LPS) [43]. In that study, they performed a microarray analysis and compared this to the control, and found that 15 miRNAs were dysregulated in the LPS exposed group. As LPS is a component of the outer membrane of Gram-negative bacteria, they hypothesized that the HMECs cells sensitization by LPS may result in differentially expressed miRNAs patterns. The study was the first to demonstrate the difference in miRNAs expression in HMECs after an inflammatory stimulus. In the predicted target genes of the altered miRNAs were the genes responsible for the innate inflammatory response, cell growth, cell differentiation, communication and adhesion or complement activation, so the authors of the study hypothesized that altered miRNAs may take part in triggering the acute inflammation response [43].

Another study presented a link between miRNA expression and Toll-like receptor (TLR) signaling activation in response to an inflammatory stimulus [33]. TLR has a well-documented role in OM development due to bacterial exposition, however, whether miRNAs were part of this signaling pathway was never elucidated [44]. Bacteria, such as *Haemophilus influenzae*, *Streptococcus pneumoniae*, and *Moraxella catarrhalis* are responsible for most cases of acute OM [45]. TLRs are triggered by these pathogens and play an important role in activating an inflammation response, as they evoke changes in the cytokines gene expression, resulting in mucosal hyperplasia and leukotic infiltration in the middle ear [46]. Samuels et al. demonstrated that proinflammatory cytokines, such as tumor necrosis factor-alpha (TNF-α) and interleukin-1 beta (IL-1 β) induced miRNA-146 expression in HMECs [33]. Moreover, increased miRNA-146 expression in OM patients correlated with increased middle ear epithelial thickness when compared to the control, showing miRNA-146 might be involved in stimulating mucosal hyperplasia [33]. Other studies supported this thesis, showing the potential of miRNA-146 in triggering acute inflammation via enhancing the TLR signaling pathway [47].

A study conducted by Val et al. was the first to confirm the presence of miRNAs in the exosomes in the middle ear fluid of patients suffering from otitis media with effusion (OME) [37]. From 800 miRNAs probed in the study, the expression of 29 was significantly altered, 17 of them were unique for middle ear exosomes and miRNA-223-3p expression was the most upregulated. The target genes analysis was then conducted and showed that miRNAs found in the middle ear exosomes influence IL-8 production which, among others, takes part in neutrophil degranulation [38]. This finding supports the hypothesis that the progression from acute to chronic OM might depend on the proinflammatory mediators released after the bacterial stimulation of the middle ear epithelium [48]. Moreover, Val et al. investigated miRNAs expression in HMEECs secretions after exposition to Nontypeable *Haemophilus influenzae* (NTHi) lysate [37]. They found five miRNAs altered by the NTHi lysate and the target gene analysis showed those miRNAs potential in initiating an innate immune response. NTHi is a common pathogen in AOM etiology that activates cytokines secretion in HMECs through TLR-2 and MAPK (mitogen-activated protein kinase) pathways [39]. Perhaps a cascade of events after NTHi exposition, initiated and regulated by altered miRNA expression, may cause middle ear epithelium remodeling, enabling it to produce mucins, which consequently is responsible for the progression from AOM to the chronic form [4].

In 2021, Zhang et al. made an attempt to identify a role of miR-210 in OME. They revealed significantly lower levels of miR-210 in the serum and middle ear effusions of patients with OME compared to healthy individuals [36]. The expression levels of IL-1β, IL-6, IL-8 and TNF-a were significantly higher both in the serum and in the middle ear effusion in children with OME, whereas the levels of the IL-10 and IL-5 were lower. Moreover, nitric oxide (NO) and vascular endothelial growth factor (VEGF) levels were also higher in the OME group. Western blot confirmed the increased phosphorylation of NF-κB p65 in middle ear effusion samples from patients with OME. In vitro tests proved that overexpression of miR-210 significantly improved cell viability in LPS-treated HMEEC cells. Additionally, the reduced cell apoptosis ratio in LPS-treated HMEEC cells compared with miR-210 NC overexpression was noticed. The authors assumed and then proved that hypoxia-inducible factor-1 alpha (HIF-1α) may be a potential target gene of miR-210 [36]. Acting as an oxygen-dependent transcriptional activator, HIF-1α affects the innate immunity, proinflammatory gene expression, antibacterial activities, and cell migration [49,50].

A nuclear-enriched abundant transcript 1 (NEAT1) is a lncRNA known from various functions in different physiological and pathological processes [40]. Hu et al. decided to explore the potential role of NEAT1 in COME. The authors revealed that miR-495 was a target of NEAT1 and identified the activation of the p38 MAPK signaling pathway. This particular lncRNA could promote inflammation by influencing the cytokine levels, cell proliferation, and apoptosis [40].

### 2.2. Noncoding RNA and Cholesteatoma

#### 2.2.1. Research Directions

The first attempts to identify the underlying molecular mechanisms of cholesteatoma were directed at the mechanisms dependent on growth factors and inflammatory mediators [51,52,53,54]. It was also agreed that AMEC may develop secondary to infection and chronic inflammation and be maintained by persistent cytokine production [5]. Keratinocytes were found to secrete proinflammatory cytokines such as interleukin (IL)-1b, IL-6, and IL-8; these cytokines stimulate perimatrix fibroblasts to release other cytokines and proinflammatory factors, causing persistent inflammation in the middle ear, which might eventually lead to keratinocyte hyperproliferation and cholesteatoma development [55,56].

Epigenetic regulatory mechanisms have been a dominant theme of molecular studies in recent years. The dysregulation of small noncoding RNAs (miRNAs) has been revealed to be crucial in AMEC formation and development [9,10,11,13,14].

Many researchers have aimed to identify the mRNA targets of miRNAs in order to explain their regulatory mechanisms [28,57]. Nevertheless, the biological phenotype of cholesteatoma cells is difficult to predict, as an individual miRNA is able to control the expression of multiple mRNAs, and a particular mRNA can be regulated by numerous miRNAs [44]. Moreover, the level of mRNA expression is not strictly related to the level of protein expression [58].

The most recent studies have been dominated by a specific subgroup of noncoding RNA, circular RNA (circRNA). CircRNA has a covalescent closed-loop structure and represents a group of noncoding RNAs. Its unique structure is thought to be responsible for its significantly higher stability compared with linear transcripts, preventing circRNAs from undergoing exonucleolytic decomposition [59,60,61]. There are numerous miRNA binding sites—called miRNA response elements (MREs)—on circRNAs, which serve as competitive endogenous RNAs (ceRNAs) [13,14]. All RNA transcripts that share common MREs (circRNAs, long noncoding RNAs, pseudogenes, and protein-coding genes) can function as ceRNAs [29,59,62]. CeRNAs are responsible for ‘sponging’, which is the process of modulating miRNA activity. It is accomplished by the capture of miRNA molecules, thus, limiting their ability to influence mRNA [63]. This attribute of ceRNA is also called a ‘molecular sponge effect’, and it affects the whole network of interactions composed of ceRNAs, miRNAs, and mRNAs and, ultimately, the expression of multiple genes [63].

#### 2.2.2. miRNA

The first study to suggest the possible role of miRNAs in cholesteatoma formation was published in 2009 by Friedland et al. [28]. They found that miRNA-21 expression is upregulated in cholesteatoma tissue compared with normal skin, and that this correlates with decreased levels of PTEN (phosphatase and tensin homolog). PTEN downregulation was previously implicated in carcinogenesis, where it is also negatively regulated by miRNA-21, among other miRNAs [64].

In the abovementioned study, a model was proposed in which the lipopolysaccharide receptor was activated during middle ear infections. Its activation stimulated IL-6 secretion, which in turn resulted in the JAK/STAT/MAPK pathway activation, eventually leading to STAT3 activation and increased miR-21 expression. In effect, miR-21 upregulation results in the inhibited translation of PTEN and PDCD4 (programmed cell death 4), which are important regulators of cell division, in effect causing constant keratinocyte proliferation and cholesteatoma invasion [28].

The results of another study, conducted by Chen at al. [23], are in accordance with the findings of Friedland et al. [28]. They compared the miR-21 expression levels of cholesteatoma tissue between adults and children, observing significantly higher levels of miR-21 and decreased levels of the abovementioned targets—PTEN and PDCD4—in pediatric patients rather than in adults. This reflects the clinical observation that cholesteatoma seems more aggressive and grows faster and more expansively in children [23]. They also found that another miRNA, miR-let-7a, is upregulated in cholesteatoma cells. MiR-let-7a is a known tumor suppressor that has been shown to be deregulated in many human cancers. Its low expression might be correlated with poor treatment outcomes in head and neck squamous cell carcinoma [65]. MiR-let-7a targets HMGA2 (high-mobility group AT-hook 2 protein), an oncogene that controls proliferation, invasion, and cell apoptosis and has recently been described as a prognostic factor of tumor grade in some cancers [66].

A study conducted by Zhang et al. seems to confirm the role of miR-let-7a in the development of cholesteatoma [24]. This study documented that miR-let-7a inhibits keratinocyte proliferation through two separate mechanisms: enhancing apoptosis and causing cell cycle arrest. Moreover, their study demonstrated the relationship between miR-21 and miR-let-7a, showing that miR-let-7a inhibits the expression of miR-21 and, therefore, its target genes. As a result, miR-let-7a upregulation may increase the expression of the miR-21 target genes PTEN and PDCD4 as part of a mechanism for inhibiting keratinocyte proliferation and cholesteatoma growth [24].

This theory is supported by Chen et al., who also hypothesized that there is a relationship between the expression of miR-21 and miR-let-7a and indicated that increased levels of miR-let-7a induce miR-21 downregulation [57].

Yao et al. analyzed the role of microRNA-199a (miR-199a), a cancer-promoting microRNA, as a probable factor influencing the development of cholesteatoma [34]. They revealed that miR-199a is upregulated in cholesteatoma tissues, which is thought to promote keratinocyte proliferation, migration, and invasion in this pathology. PNRC1 was found to be a direct target of miR-199a, with a negative regulatory interrelationship [34].

Another miRNA shown to be involved in cholesteatoma pathogenesis is miR-802. According to a study conducted by Li et al., miR-802 targets more than twenty-two genes in seventeen types of tumors through at least five different pathways [41]. Although downregulated in many neoplastic tumors, miR-802 is upregulated in cholesteatoma. Its overexpression stimulates keratinocyte proliferation and cholesteatoma growth. MiR-802 targets PTEN; its upregulation is correlated with PTEN downregulation, which alleviates the inhibitory effect of PTEN on the PI3K/AKT pathway [41]. Moreover, miR-802 expression is induced by NF-kB (nuclear factor kappa-light-chain-enhancer of activated B cells), a protein complex that controls the cytokine production in response to inflammatory stimuli. This finding supports the proposal that cholesteatoma growth might be induced by cytokines secreted during the inflammation response, and that the NF-kb/miR-802/PTEN signaling pathway might play a role in expansion [67].

Zang et al. revealed that miR-203a influences keratinocyte hyperproliferation [35]. The authors demonstrated that miR-203a is downregulated in cholesteatoma keratinocytes and its target gene—Bmi1—is upregulated. Bmi1 is a complex protein that has previously been described to play a role in carcinogenesis. Bmi1 enhances the expression of p-Akt, which contributes to cell hyperproliferation and invasion and, in effect, plays a role in tumor growth and metastasis [68]. Therefore, another signaling pathway model was proposed by the authors, suggesting that the miR-203a/Bmi1/p-Akt axis might participate in the development of cholesteatoma and become a future target in pharmacological treatment [68].

A study by Sui et al. confirmed the negative correlation between miR-142-5p and cyclin-dependent kinase 5 (CDK5) in terms of expression and regulation [32]. CDK5 was previously reported to contribute to increasing inflammatory responses [69,70]. Sui et al. observed CDK5 overexpression in cholesteatoma tissue samples, which was found to be positively correlated with levels of the cytokines TNF-α, TGF-β1, IL-5, IL-6, and IL-17A [32]. It is commonly known that TNF-α, IL-6, and IL-17A affect the development and proliferation of epidermal cells [71]. Therefore, it was suspected that the formation of a specific inflammatory microenvironment in cholesteatoma may result from the upregulation of IL-5 in AMEC [32]. Moreover, Sui et al. hypothesized that the CDK5/MAPK/NF-κB pathway is used by miR-142-5p to regulate the secretion of inflammatory cytokines [32].

Previous studies have highlighted the role of new blood vessel formation in the development of tissue masses such as cholesteatoma. Increased angiogenesis is essential in tumor growth and expansion, as it enables nutrient delivery and oxygen exchange [72]. Li et al. investigated the presence of miRNAs in exosomes derived from cholesteatoma perimatrix fibroblasts (hCPFs-Exo) and found that miR-106b-5p is downregulated [31]. As miR-106b-5p inhibits the expression of angiopoietin 2 (Angpt2), a growth factor that regulates new vessel formation, they hypothesized that its downregulation promotes angiogenesis by disinhibiting Angpt2 [31].

The results of a microarray analysis of miRNAs expressed in cholesteatoma demonstrated that the expression of 219 miRNAs is altered (22 upregulated and 175 downregulated) when compared with normal skin. The results for some miRNAs were similar to those of previous studies, such as in regard to the upregulation of miR-21-3p, whereas they were divergent for others. Surprisingly, in that study, the expression of members of the miR-let-7a family did not differ in cholesteatoma compared with normal skin. The GO and KEGG pathway target gene analyses were performed on five selected miRNAs and showed that these miRNAs take part in proliferation, differentiation, the cell cycle, and apoptosis [26].

Recent evidence shows that miRNAs are also involved in osteoclast formation in cholesteatoma tissue. Gong et al. established that miR-17 is secreted by keratinocytes and—through exosomes—can interact with fibroblasts, favoring their differentiation into osteoclasts by upregulating the RANKL levels [27]. RANKL, a protein that is a member of the TNF (tumor necrosis factor) family, has been reported to participate in osteoclast activation [73]. The authors hypothesized that, upon osteoclast hyperactivation, the balance between bone formation and resorption is lost, which may support further cholesteatoma development.

Taken together, the results of recent studies imply that miRNAs might be crucial regulators of post-translational processes, leading to a series of events that, in effect, result in cholesteatoma progression. There is evidence that they participate in almost every stage of cholesteatoma formation, as they are engaged in keratinocyte hyperproliferation, angiogenesis, and fibroblast-to-osteoclast transformation.

#### 2.2.3. circRNA

In 2018, a study on cholesteatoma formation conducted by Gao et al. revealed that long noncoding RNAs can serve as ceRNAs [29]. However, the real breakthrough was their study published in 2020, using microarray analysis and confirmation by qRT-PCR [25]. The authors determined that the profiles of circRNAs in AMEC were divergent from those derived from normal skin samples. Using a bioinformatics approach, the authors constructed the network composed of circRNA–lncRNA–miRNA–mRNA and investigated the ‘sponging’ potential of circRNAs [25]. Two circRNAs sharing common MREs were selected (hsa-circRNA-101458 and hsa-circRNA-102747). CircRNA-102747 was found to interact with miR-21-3p [25]. The role of the miR-21 family in the formation and spread of cholesteatoma has been previously demonstrated [28,57]. The link between circRNA-101458 and miR-let-7a-3p has also been confirmed [25]. The upregulation of miR-let-7 from the same family was thought to promote the benign nature of AMEC by supporting antiproliferation [23,74]. The authors hypothesized the opposite regulatory influence on mRNAs of different circRNA–lncRNA–miRNA networks [25]. Thus, circRNA-101458/miR-let-7a-3p and circRNA-102747/lncRNA-uc001kfc.1/miR-21-3p may be, respectively, regarded as benign and malignant regulators. This suggests they may be considered as potential targets for the conventional treatment of AMEC [25].

Knowing that miR-125a-5p [75,76,77] and miR-22-3p [78,79,80] exhibit antitumor activity in some types of malignant neoplasms, Hu et al. identified that circ_0074491 plays the role of decoy for miR-125a-5p and miR-22-3p [42]. They suspected its regulative effect on AMEC progression involves miR-125a-5p and miR-22-3p. The results confirmed that the PI3K/Akt pathway is involved in the regulatory influence of circ_0074491 on the adsorption of miR-125a-5p and miR-22-3p in cholesteatoma keratinocytes [42]. Circ_0074491 upregulation was found to be negatively correlated with the malignant behaviors of AMEC keratinocytes [25]. Xie et al. demonstrated that the expression of hsa_circRNA_000319 was significantly decreased whereas hsa_circRNA_104327 and hsa_circRNA_404655 were expressed significantly higher in cholesteatoma samples [81]. The authors speculated that hsa_circRNA_404655 may play a role of the sponge for miRNA-3664-3p, influencing the proliferation and local invasion of AMEC [81]. The lower expression of miRNA-3664-3p may lead to an increased expression of p53 via growth differentiation factor 15 (GDF15) [82].

Another study presenting the circRNA–miRNA–target mRNA axis was published by Liu et al. in 2021 [6]. The authors selected miR-508-3p and hsa_circ_0000007 and identified significantly higher expression of miR-508-3p in cholesteatoma samples than in HaCaT cells or retroauricular skin. In addition, has_circ_0000007 negatively influenced the expression of miR-508-3p. Their results revealed that miR-508-3p directly targets PTEN and influences the phenotype of AMEC. The authors concluded that, when present in cholesteatoma cells miR-508-3p is able to inhibit apoptosis and promote proliferation via the PTEN/PI3K/Akt pathway [6].

#### 2.2.4. Practical Implications

An experimental study using nanotechnology conducted by Zheng et al. represents an important step forward in the practical use of miRNAs in treatment [30]. Based on AMEC tissue, the authors prepared and delivered rubine, a regulator of miR-34a, into cells. Their experiments revealed that miR-34a may target EGFR and, via this mechanism, suppress the growth and development of tumors. Overexpression of miR-34a can promote apoptosis and prevent cell proliferation and migration. The study thus confirmed that the expression of miR-34 may be enhanced by nanoparticles and thereby influence CyclinD1, Bcl-2, and Cdk6 [30].

## 3. Conclusions

In this comprehensive review, we summarized the contemporary molecular biology achievements analyzing the modulating role of noncoding RNAs and their potential mechanisms in the pathogenesis of different forms of otits media and especially chronic otits media with cholesteatoma. The recent progress that has been made in biomolecular research indicates the direction for future studies, wherein the circRNA–lncRNA–miRNA–mRNA axis is likely to be key. Both in vitro and in vivo studies on the regulatory influence of ceRNA on circRNAs are necessary for progress in elucidating the pathogenesis of AOM, rAOM, COME and AMEC. This may ultimately result in a paradigm shift in cholesteatoma treatment modalities, evolving into pharmacotherapy based on the modulation of transcriptional control.

## Figures and Tables

**Table 1 ijms-24-06752-t001:** Identification of publications meeting the inclusion criteria based on the results of searching the PubMed, Web of Science and Scopus databases.

	Database	PubMed	Scopus	Web of Science
Searched Terms				
‘otits media’ and ‘miRNA’		4	4	4
‘otits media’ and ‘circRNA’		0	0	0
‘cholesteatoma’ and ‘miRNA’		16	13	11
‘cholesteatoma’ and ‘circRNA’		3	4	5
Total articles identified *		21	19	18
Total articles included *		27		

* duplications removed.

**Table 2 ijms-24-06752-t002:** Selected noncoding RNAs and their potential role in AOM, rAOM, COME and cholesteatoma.

Noncoding RNA	Regulatory Mechanism or Signaling Pathway	Influence	References
miR-let-7a	HMAG2	proliferation of kerationocytes	[23,24,25]
miR-10a-5p	PI3K/Akt	hyperproliferation of cholesteatoma	[26]
miR-16-1-3p	PI3K/Akt	hyperproliferation of cholesteatoma	[26]
miR-17	RANKL	differentiation of fibroblasts into osteoclasts, osteoclasts activation	[27]
miR-21	PTEN, PDCD4	proliferation of kerationocytes	[23,25,28,29]
miR-34a	PTEN/PI3K/Akt/EGFR	proliferation, apoptosis	[30]
miR-106b-5p	Angpt2	angiogenesis	[31]
miR-142-5p	CDK5/MAPK/NF-κB	positive correlation with: TNF-α, TGF-β1, IL-5, IL-6, and IL-17A	[32]
miR-146	(TNF-α) (IL-1 β)/TLR	mucosal hyperplasia	[33]
miR-199a	PNRC1	keratinocyte proliferation, migration, and invasion	[34]
miR-203a	Bmi1/p-Akt	proliferation, migration and antiapoptotic abilities	[35]
miR-210	NF-κB/HIF-1α	cell viability, apoptosis	[36]
miRNA-223-3p	IL-8	neutrophil degranulation	[37,38]
miR-378a-3p + miR-378i, miR-200a-3p, miR-378g, miR30d-5p, miR-222-3p miRNAs	TLR-2 and MAPK	epithelium remodeling	[37,39]
NEAT1/miR-495	p38 MAPK	cell proliferation, levels of inflammatory cytokines, cell apoptosis	[40]
miR-802	PTEN/PI3K/Akt	proliferation of kerationocytes	[41]
circ-0000007/ miR-508-3p	PTEN/PI3K/Akt	proliferation apoptosis	[20]
circ-0074491/ miR-22-3p	PI3K/Akt	proliferation, colony formation, apoptosis of keratinocytes	[42]
circ-0074491/ miR-125a-5p	PI3K/Akt	proliferation, colony formation, apoptosis of keratinocytes	[42]
circRNA-101458/ miR-let-7a	PI3K/Akt	proliferation of kerationocytes	[25]
circ-102747/ miR-21-3p	PI3K/Akt	formation and invasion of cholesteatoma	[25]

## Data Availability

Data available from the corresponding author on request.

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
