# Peer review of "Micro RNAs and Circular RNAs in Different Forms of Otitis Media"

_ijms, 2023, doi:10.3390/ijms24076752_

Round 1
Reviewer 1 Report
The authors have reviewed the current advance on the functions of microRNAs and circular RNAs (miRNAs) in acute, recurrent, and chronic forms of otitis media. I believe that the work is comprehensive and should make significant contribution to the relevant fields. I have two major technical concerns as I have listed below, mainly the organization of Introduction and the inadequate presentation of Materials and Methods. I believe these suggestions could help improve the overall presentation of this manuscript.
Title:
Change “noncoding RNA” to “MicroRNAs and circular RNAs” because the review only covers these two types of noncoding RNAs but not other types of noncoding RNAs
Abstract:
Line 12, to better understanding “of” the…
Line 15, delete “(MeSH)”
Introduction:
I believe it would be appropriate to introduce the medical problems first, then explain the importance of microRNAs. Therefore, I would recommend that the authors re-organize the structure of Introduction to reflect this order.
Results:
Table 1: the title is oversimplified…more details are needed; also, change “Table 1”to plain text but not italicized…same problem for the format of the title of “3.2.3. circRNA” (line 310)
Materials and Methods:
Apparently, a lot more details, step by step, are needed so that the search could be repeated by the readers, e.g., “The inclusion and exclusion criteria were all defined…” bit what are these criteria???
Author Response
Responce to Reviewer #
General information: I would like to thank the Reviewer for the revision.
Reviewer’s comment: The authors have reviewed the current advance on the functions of microRNAs and circular RNAs (miRNAs) in acute, recurrent, and chronic forms of otitis media. I believe that the work is comprehensive and should make significant contribution to the relevant fields. I have two major technical concerns as I have listed below, mainly the organization of Introduction and the inadequate presentation of Materials and Methods. I believe these suggestions could help improve the overall presentation of this manuscript.
Reply: I appreciate Your assessment of the substantive value of the study and Your suggestions that contributed to the improvement of this manuscript.
Reviewer’s comment: Title: Change “noncoding RNA” to “MicroRNAs and circular RNAs” because the review only covers these two types of noncoding RNAs but not other types of noncoding RNAs.
Reply and action: Thank You for this comment. I fully agree that the revised title is more precise. The title has been changed.
Reviewer’s comment: Abstract: Line 12, to better understanding “of” the…
Reply and action: Thank You for pointing out this oversight. The sentence has been corrected.
Reviewer’s comment: Abstract: Line 15, delete “(MeSH)”
Reply and action: Thank You for this comment. The sentence has been corrected.
Reviewer’s comment: Introduction: I believe it would be appropriate to introduce the medical problems first, then explain the importance of microRNAs. Therefore, I would recommend that the authors re-organize the structure of Introduction to reflect this order.
Reply and action: Thank You for this valuable comment. The Introduction section has been re-organized according to the reviewer’s suggestion.
Reviewer’s comment: Results: Table 1: the title is oversimplified…more details are needed; also, change “Table 1”to plain text but not italicized…same problem for the format of the title of “3.2.3. circRNA” (line 310)
Reply and action: Thank You for these comments. The title of the Table 1 has been changed into more detailed version. The italics in the mentioned lines have been changed into plain text.
Reviewer’s comment: Materials and Methods: Apparently, a lot more details, step by step, are needed so that the search could be repeated by the readers, e.g., “The inclusion and exclusion criteria were all defined…” bit what are these criteria???
Reply and action: Thank You for these valuable comments. The inclusion end exlusion criteria have been presented more rigorously and transparently. Step by step, the procedure for selecting scientific articles qualified for the final analysis was presented.
Nevertheless, the structure of the manuscript has been changed according to the claim of one of the Reviewers. I was obliged to modify the structure of the manuscript to ‘fit the basic consideration of a review article’. Therefore, the results and Material and methods sections were included into Literature review section. Table 2and Table 3 were fused.
I belive these changes have no negative influence on the content of the manuscript.
Reviewer 2 Report
I had gone through the present manuscript and my comments were attached herein. This article was aimed to present a complete overview of the latest progress on the roles of miRNA and circRNA in AOM, rAOM, OME, and AMEC, based on an extensive literature review covered from 2009 to 2022. The review target is very interesting and a very hot topic in the recent years. However, the aim and scope of the present manuscript was a little limited. In addition, the structure of this manuscript did not fall into the common skeleton of a review article. In summary, this manuscript may meet the minimum criteria of this journal and is not recommended to accept for publication in International Journal of Molecular Sciences in the present form. The acceptance of this manuscript should be reconsidered after major revisions addressed as following.
1. The manuscript was written well, however, there were still some minor typographical, grammar and format errors presented. For example, the names of bacteria in lines 126-127 should be in italics. Authors have to check and revise these errors carefully.
2. The main problem was that a review article did not need “Results” and “Materials and Methods” section. Authors have to modify the structure of the present manuscript to fit the basic consideration of a review article. Moreover, lines 13-16, this information would usually not appear in the abstract.
3. Although Tables 2 and 3 provides different information, it is helpful for the readers to combine these two tables into one.
4. If possible, authors have to expand the scope of this manuscript to some more related targets.
5. In the References section, the writing manner of several references did not follow the style of this journal. Authors have to check and revise these errors carefully, including refs 16, 26, 34, 35, 58, 61, 65, 69, 74, 82, and 83, but not limited.
Author Response
Response to Reviewer #2
General information: I would like to thank the Reviewer for the revision.
Reviewer’s comment: I had gone through the present manuscript and my comments were attached herein. This article was aimed to present a complete overview of the latest progress on the roles of miRNA and circRNA in AOM, rAOM, OME, and AMEC, based on an extensive literature review covered from 2009 to 2022. The review target is very interesting and a very hot topic in the recent years. However, the aim and scope of the present manuscript was a little limited.
Reply: Thank You for appreciating that the review target is very interesting. As You highlighted, we aimed to present a complete literature overview of the current knowledge on the miRNAs and circRNAs in acute, chronic and recurrent inflammatory pathologies of the middle ear. To the best of our knowledge, we analysed all the available literature concerning miRNAs and circRNAs in different forms of otitis media. Moreover, we intended to present the major controlling mechanisms and pathways as well as their potential or proven clinilal result. Although there is a urgent need to search for mechanisms controlling the development of inflammatory diseases of the middle ear and non-surgical methods of their treatment, there is a paucity of publications on this subject. We agree that even more detailed analysis is desired. Therefore we are working on research projects to present our own results focused not only on molecular but also clinical effects. We will be greatful for all Your hints and comments regarding the aim and scope of this study and the ongoing research on the issue.
Reviewer’s comment: In addition, the structure of this manuscript did not fall into the common skeleton of a review article. In summary, this manuscript may meet the minimum criteria of this journal and is not recommended to accept for publication in International Journal of Molecular Sciences in the present form. The acceptance of this manuscript should be reconsidered after major revisions addressed as following.
Reply and action: In the following sections of this response we present and explain all the changes and corrections that have been made in accordance with the Reviewer’s suggestions.
Reviewer’s comment: The manuscript was written well, however, there were still some minor typographical, grammar and format errors presented. For example, the names of bacteria in lines 126-127 should be in italics. Authors have to check and revise these errors carefully.
Reply and action: It is glad to hear Your opinion that the manuscript was written well. Thank You very much. We thoroughly checked the manuscript and corrected identified errors. We believe that these minor errors do not influence on the substantive value of the manuscript.
Reviewer’s comment: The main problem was that a review article did not need “Results” and “Materials and Methods” section. Authors have to modify the structure of the present manuscript to fit the basic consideration of a review article.
Reply and action: Thank You for Your opinion. We strongly agree that the typical review article doesn’t need the sections characteristic to the research paper. Nevertheless, we thought it would be more clear if organized in the previously proposed way. According to Your suggestion and to meet the requirement of IJMS, we have modified the structure of the manuscript. The data from the Material and methods and Results sections were incorporated to the Literature review section. It is a consequence of the other Reviewer’s claim to keep all the data needed so that the search could be repeated by the readers.
Reviewer’s comment: Moreover, lines 13-16, this information would usually not appear in the abstract.
Reply and action: The sentence from the lines 13-16 has been removed.
Reviewer’s comment: Although Tables 2 and 3 provides different information, it is helpful for the readers to combine these two tables into one.
Reply and action: Thank You for this comment. Tabel 2 and 3 have been fused into one. It is far more clear in this form.
Reviewer’s comment: If possible, authors have to expand the scope of this manuscript to some more related targets.
Reply: Thank You for Your interesting opinion. Indeed, we had considered expanding the scope of the manuscript at the initial stage of this study and presenting more detailed clinical outcomes of noncoding RNAs releted mechanisms in the middle ear but we resigned in the end. We sustained the initial decision that the manuscript should present the overview of the studies on the subject of miRNAs and circRNAs to indicate the potential directions in desingning of further studies and to act as an incentive for other researchers. If we had added all the clinical aspects of inflammatory processes in the middle ear, the study would have been too extensive, multithreded and the potential readers would have been confused. A few years ago we initiated projects focused on clinical aspects of selected noncoding RNAs regulatory functions in various forms of otitis media and published our results (Adamczyk P, Narożna B, Szczepankiewicz A, Bręborowicz A, Pucher B, Kotowski M, Sroczyński J, Kałużna-Młynarczyk A, Szydłowski J. Decreased miRNA-320e correlates with allergy in children with otitis media with effusion. Auris Nasus Larynx. 2021;48(6):1061-1066. doi: 10.1016/j.anl.2021.03.003). The studies concerning the influence of noncoding RNAs on the agressivness and recurrence of cholesteatoma, the persistence of otitis media with effusion, sudden sensorineural deafness etc. are still in progress. We hope to be able to preset their results in the nearest future.
Reviewer’s comment: In the References section, the writing manner of several references did not follow the style of this journal. Authors have to check and revise these errors carefully, including refs 16, 26, 34, 35, 58, 61, 65, 69, 74, 82, and 83, but not limited.
Reply & Action: Thank You for this comment. All the references have been checked and corrected if necessary.
Round 2
Reviewer 1 Report
I appreciate very much the efforts that the authors have devoted to improving their manuscript. I have no more questions.
Author Response
Responce to Reviewer #1
Reviewer’s comment: I appreciate very much the efforts that the authors have devoted to improving their manuscript. I have no more questions.
Reply: Thank You very much.
Reviewer 2 Report
I had gone through the revised manuscript and author had provided more solid responses as compared with the previous version. I felt that the present manuscript may meet the criteria of International Journal of Molecular Sciences and this manuscript could be recommended to accept for publication after some grammar and format checks.
Author Response
Response to Reviewer #2
Reviewer’s comment: I had gone through the revised manuscript and author had provided more solid responses as compared with the previous version. I felt that the present manuscript may meet the criteria of International Journal of Molecular Sciences and this manuscript could be recommended to accept for publication after some grammar and format checks.
Reply: Thank You very much for Your opinion. The manuscript was checked by the English native speaker and some minor corrections were made.